# HIV Cerebrospinal Fluid Escape and Neurocognitive Pathology in the Era of Combined Antiretroviral Therapy: What Lies Beneath the Tip of the Iceberg in Sub-Saharan Africa?

**DOI:** 10.3390/brainsci8100190

**Published:** 2018-10-20

**Authors:** Dami Aderonke Collier, Lewis Haddow, Jay Brijkumar, Mahomed-Yunus S. Moosa, Laura Benjamin, Ravindra K. Gupta

**Affiliations:** 1Division of Infection and Immunity, University College London, London WC1E 6BT, UK; ravindra.gupta@ucl.ac.uk; 2Africa Health Research Institute, KwaZulu-Natal Durban 4001, South Africa; 3Institute for Global Health, University College London, London WC1E 6BT, UK; lewis.haddow@ucl.ac.uk; 4RK Khan Hospital, KwaZulu-Natal Durban 4092, South Africa; jbrijkumar@gmail.com; 5Department of Infectious Diseases, University of KwaZulu-Natal, Durban 4041, South Africa; Moosay@ukzn.ac.za; 6Institute of Infection and Global Health, University of Liverpool, Liverpool L69 3BX, UK; L.Benjamin@liverpool.ac.uk

**Keywords:** HIV, central nervous system, escape, discordance, independent replication, resistance

## Abstract

Neurocognitive impairment remains an important HIV-associated comorbidity despite combination antiretroviral therapy (ART). Since the advent of ART, the spectrum of HIV-associated neurocognitive disorder (HAND) has shifted from the most severe form to milder forms. Independent replication of HIV in the central nervous system despite ART, so-called cerebrospinal fluid (CSF) escape is now recognised in the context of individuals with a reconstituted immune system. This review describes the global prevalence and clinical spectrum of CSF escape, it role in the pathogenesis of HAND and current advances in the diagnosis and management. It highlights gaps in knowledge in sub-Saharan Africa where the HIV burden is greatest and discusses the implications for this region in the context of the global HIV treatment scale up.

## 1. Introduction

Globally, 36.7 million people are living with HIV with 21 million of these on antiretroviral (ART) [1]. Before the advent of ART, HIV-associated central nervous system (CNS) pathology included opportunistic infections such as cryptococcal meningitis, toxoplasmosis, tuberculosis meningitis and cytomegalovirus encephalitis, CNS Lymphoma but also HIV encephalitis. Since the rollout of ART, the incidence of opportunistic infections and primary CNS lymphoma have fallen along with a reduction in HIV-associated mortality [2]. However, HIV-associated neurocognitive disorder (HAND) remains an important problem.

A contentious issue is whether independently replicating HIV in the brain parenchyma and the proxy for this—the presence of HIV virus in the cerebrospinal fluid (CSF), is responsible for neurocognitive impairment (NCI) in the post-ART era. The terms commonly used are CSF discordance and CSF escape. The definitions of these terms are based on the quantification of CSF HIV RNA in paired CSF and plasma specimens (see Table 1). CSF discordance is defined as CSF viral load (VL) greater than 0.5 or 1log_10_ of the plasma VL and CSF escape is defined as any VL in the CSF above the limit of detection of the assay used (usually 50 copies/mL) when the VL in the plasma is undetectable by the same assay [3,4,5]. Although CSF HIV VL does correlate well with brain HIV RNA levels [6], the causal relationship between CSF escape/discordance and HAND is unclear. This leads to questions about the utility of CSF HIV RNA as a diagnostic tool for HAND and what level of CSF HIV RNA is considered to be problematic.

In this review, we describe (1) the prevalence of HAND globally, (2) the clinical spectrum of CSF escape/discordance (3) the contribution CSF escape/discordance to HIV-associated neurocognitive disorder (4) critically review the role of CSF HIV RNA and the evolution of HIV drug resistance in the pathogenesis of CSF escape/discordance, (5) appraise the current state of clinical biomarkers in the diagnosis of HAND and CSF escape/discordance and (6) review progress with management.

It is worth noting that the majority of research on HAND has been done in Western settings where Subtype B HIV-1 predominates but little is known about HAND in sub-Saharan Africa (SSA) where subtypes C, A, D, G and AG predominate. We aim to identify the research gaps globally but highlight the implications in SSA where the burden of HIV is greatest.

## 2. Prevalence of HAND in the Post-ART Era

It is clear that since the advent of ART, the prevalence of the most severe form of HAND–HIV associated dementia (HAD), has markedly reduced [7]. The Multicentre AIDS Cohort Study (MACS), a USA based study that observed the natural progression of HIV disease, found the incidence of HAD was 22.3 per 1000 person years in the pre-ART era from 1990 to 1995 [2]. Following the introduction of ART, the incidence halved to 11.9 per 1000 person years between 1996 and 1998 [2]. However, the milder forms of HAND; mild neurocognitive disorder (MND) and asymptomatic neurocognitive impairment (ANI) are increasingly recognised in patients treated with ART [8,9]. The CNS HIV Antiretroviral Therapy Effects Research (CHARTER) study, another USA based study initiated after the advent of ART, had 85% of participants on treatment and showed that the spectrum of HAND has shifted, such that up to 50% of the cohort had HAND but of the least severe type [8]. Similarly, a study in Switzerland conducted on patients with durable plasma virological suppression, with and without a cognitive complaints had a prevalence of HAND of 84% and 64% respectively. The proportion of these with ANI were 24% and 60% respectively [9].

There is evidence to suggest that HIV-1 subtype plays a role in neurocognitive dysfunction. Neurocognitive impairment has been associated with CRF_02AG when compared with subtype G in Nigeria [10]. In Uganda, a study found that participants with subtype D virus were at greater risk of HAD compared to patients with subtype A virus [11]. It is therefore important to know if there are regional variations in the prevalence of HAND particularly in SSA where the global burden of HIV-1 is greatest and where sub-type C virus predominates. The diagnosis of HAND is based on the Frascati Criteria which requires the use of neuropsychological (NP) tests to assess cognitive domains and determine the extent of decline in performance compared to a normative population together with an assessment of functional decline (see Table 2) [12]. The prevalence of HAND in sub-Saharan Africa (SSA) is uncertain because the standard NP tests used to assess cognitive domains require specific training to deliver and sometimes specialised equipment. In addition, these tests were developed in resource rich settings and therefore limited by their cultural appropriateness for SSA. The lack of neurocognitive normative comparison data for this setting is also a limitation [13]. These challenges have been partly addressed by various diagnostic accuracy studies and the collection of neurocognitive normative data [14,15,16,17,18]. In the absence of NP testing, several studies in SSA have attempted to determine an approximate prevalence of HAND using screening tools. One such screening tool is the international HIV dementia scale (IHDS), which has fair to moderate sensitivity (45%) and specificity (79%) in South Africa at a cut off of <11 [14]. In a study in Uganda the sensitivity and specificity were 80% and 55% respectively [19]. A systematic review of studies using the IHDS to measure the prevalence of HAND in SSA, amongst patients on ART for at least 6 months, reported a pooled prevalence of 30.4% (range 4.8% to 80%) [20]. The wide range of prevalence estimates reflect the heterogeneity of the studies included in the systematic review. Few of the studies excluded patients with psychiatric presentations, substance abuse, opportunistic infection and none used neuroimaging to assess for an alternative cause of NCI [20]. This raises questions about diagnostic overestimation. A prevalence study of HAND in South Africa, using NP testing in treatment naïve persons found a prevalence of 76% (9% ANI, 42% MND) [21]. Two studies have examined the prevalence of HAND using NP testing in ART treated persons in South Africa and Malawi and found a prevalence of 66% (54% MND) [22] and 70% (55% ANI, 12% MND) [17] respectively.

There is controversy around the actual prevalence of HAND because based on these criteria, the prevalence of HAND is sufficiently high to warrant questions about diagnostic overestimation [23,24].

## 3. The Spectrum of CSF Escape/Discordance

### 3.1. Prevalence of CSF Escape/Discordance

In the pre-ART era, HIV encephalitis was common and associated with high CSF viral load [25]. Since the advent of ART, peripheral suppression of HIV-1 has largely been accompanied by suppression in the CNS. However, this is not the case in every treated individual and the occurrence of CSF escape/discordance is now recognised [3]. Eden showed in a cross-sectional study of 69 asymptomatic, virally suppressed patients on ART that the prevalence of CSF escape was 10% [26]. This was followed up by a retrospective cohort of 75 neurologically asymptomatic patients who had been peripherally suppressed with an HIV-1 load of <50 copies/mL for a median of 53 months [27]. They found that 36% of these participants had CSF HIV-1 VL above the lower limit of quantification (20 copies/mL) on at least one occasion. However repeatedly elevated CSF HIV-1 load was rare (3%) and so the majority were considered to be CSF blips. The true prevalence of CSF escape in virologically suppressed persons is uncertain as the prevalence is influenced by the limit of detection of the HIV viral load assay and the CSF sample volume used. Studies that have used a higher detection limit have reported a lower prevalence of CSF escape [28,29]. In the Comorbidity in Relation to AIDS (COBRA) cohort study, where all participants were peripherally suppressed, the prevalence of CSF escape was 1.5% (2/134) when the limit of detection of CSF HIV-1 was 40 copies/mL [28]. A sub study of participants from the CHARTER cohort with plasma VL < 50 copies/mL, similarly used a limit of detection of CSF HIV-1 of 40 copies/ml and estimated a prevalence of CSF escape of 1.7% (2/121) [29].

A retrospective longitudinal study of neurological symptomatic cases from a clinic in Boston found a prevalence of CSF escape of 6% (12/200) [30]. A retrospective study in the United Kingdom in HIV positive patients who underwent a clinically indicated lumbar puncture found a prevalence of CSF discordance and CSF escape of 15% (22/146) and 6% (9/146) respectively [31]. A similar study from another hospital in the United Kingdom found a prevalence of CSF escape of 11% (16/142) [32]. The PARTITION study prospectively recruited participants with low level viraemia (LLV) and patients undertaking a clinically indicated lumbar puncture and found the prevalence of CSF discordance of 18% (7/40) and 12% (13/113) respectively [33]. Amongst the National NeuroAids Tissue Consortium (NNTC) cohort followed up since 1998, 62% of whom had neurological symptoms at the time of lumbar puncture, the prevalence of CSF escape of 6.8% (26/426) [30]. Mukerji has since described a combined cohort from CHARTER, NNTC and HIV Neurobehavioural Research Centre (HNRC) with a pooled prevalence of CSF escape of 7.2% (77/1063) [34]. As of yet, there are no published prevalence studies of CSF escape/discordance from SSA, so the burden in this region remains unknown.

### 3.2. The Clinical Presentation

The spectrum of clinical manifestation of CSF escape/discordance in the ART era ranges from asymptomatic to coma and death. There have been several cases reports and case series describing patients with chronic HIV on ART with symptomatic CSF escape/discordance [3,4,35,36,37,38]. These cases present with an acute or subacute meningoencephalitis with a broad spectrum of symptoms including fever, headache, seizure, personality change, psychosis, cortical and subcortical neurological deficits, coma and death [3,4,35,36,37,38].

### 3.3. Pathogenesis 

CSF HIV RNA is thought to be a proxy for independent replication of HIV in the brain parenchyma as it correlates significantly with brain HIV RNA [6,39]. HIV in the brain is found in neurones, astrocytes and oligodendrocytes but preferentially infects microglia and perivascular macrophages [40,41], both of which express the HIV receptor CD4 and co-receptor CCR5 [42]. CSF escape/discordance is therefore thought to be a reflection of infection of the brain parenchyma. Recent discovery of a major restriction factor in macrophages, SAMHD1, prompted a rethink as to whether macrophages could indeed be infected by HIV in the face of this host defence protein, or whether observed infection was merely the result of engulfment of HIV infected T cells [43]. Subsequent studies confirmed the ability of SAMHD1 expressing macrophages to sustain HIV infection in CNS tissue [44,45], most likely explained by the deactivation of SAMHD1 by phosphorylation in sub populations of macrophages entering early stages of the cell cycle [46]. Drugs such as Etoposide and Vorinostat have been found to activate SAMDH1, thereby protecting macrophages from HIV-1 infection [47]. These long-lived brain macrophages are key to the establishment of a latent reservoir of HIV-1 infection in brain and may impact the goals of HIV-1 eradication and functional cure [48].

The pathophysiological process behind asymptomatic and symptomatic CSF escape/discordance appear to be distinct [49]. In asymptomatic CSF escape, the CSF is acellular and there are no hallmarks of neuroinflammation; biomarkers for neuronal injury are normal such as neurofilament light chain protein [26]. The source of HIV RNA is thought to be either through passive trafficking within infected immune cells from the plasma and subsequent clonal expansion or due to activation of a latent reservoir from long-lived CNS cells such as macrophages and microglia [40,41]. The presence of elevated CSF neopterin supports the later [26]. These incidences of CSF escape do not have an impact on the resistance profile nor appear to be associated with drug CNS penetration effectiveness (CPE) scores. It is unknown whether these patients go on to develop symptomatic CSF escape.

In symptomatic CSF escape, there is neuroinflammation characteristic of a meningoencephalitis, with cellular CSF predominated by CD8^+^ lymphocytes [37,50]. Brain histology supports the presence of perivascular lymphocytic infiltrates with a CD8^+^ predominance [4]. This may represent an immune response to compartmentalised antigen in the CNS and it occurs in the presence of a moderately reconstituted immune system, with a respectable CD4 count and so distinct from immune reconstitution inflammatory syndrome (IRIS) which occurs at low CD4 count [3,4,35,36,37].

### 3.4. Risk Factors for CSF Escape/Discordance

There is evidence to suggest that CSF escape/discordance in neurosymptomatic individuals or participants having a lumbar puncture for clinical reasons is associated with a longer duration of ART [26,30], low nadir CD4 [26,30,51], ART regimens consisting of a ritonavir-boosted protease inhibitor (bPI) [34], LLV in plasma [33], history of plasma vial blips [26] low CPE scores [3,4,32] and frequent drug interruptions [26].

CSF escape has also been described in patients on PI monotherapy [52]. However, in PI monotherapy drug trials, there have been very few cases reported [53,54]. In the MOST study, 5 of 42 participants on boosted lopinavir developed CSF escape [55]. In the PROTEA sub-study, no participants out of 71 whom had CSF analyses, developed CSF escape/discordance [56]. Similarly, in the PIVOT sub-study none of the 11 of 88 participants on PI monotherapy who consented to have a lumbar puncture had CSF escape [53]. In the MONOI study 2 neurosymptomatic individuals of the 112 in the PI monotherapy arm were evaluated with CSF analyses and were both found to have CSF escape [57]. In addition, participants on PI monotherapy have not been found to have worse performance on NP testing or symptomatic NCI compared to those on triple ART in clinical trials [54,58]. The interpretation of the association between CSF escape/discordance and bPI is unclear, as the rationale for bPI monotherapy use in select patients may be a confounder. Patients who are given bPI monotherapy may be poorly-adherent and bPI is used due to its high barrier to resistance [59] or may be ill patients and bPI monotherapy is used in order to minimise drug interactions [60] or may have experienced toxicity from NRTIs and bPI is used as a temporary measure [61]. Boosted PI-based therapy may represent poor adherence, extensive ART experience and the opportunity for the virus to have evolved under drug pressure. A combined cohort from CHARTER, NNTC and HNRC found a 3.1-fold increased odds of CSF escape (95% CI 1.9–5.1) on bPI-based triple therapy [34]. Some studies have not found an association between low CPE scores and CSF escape [26,31,33,62]. This may be because the value was not adjusted for drug resistance. Peluso found a median CPE score of 6.5 (range, 3–13) but when adjusted for resistance, the median adjusted CPE score was 1 (range, 0–9) [4]. Mukherji similarly found the median CPE value was 7 (interquartile range [IQR], 7–8), while resistance-adjusted CPE values were 6 (IQR 4–7) in plasma and 3.5 (IQR 3–4) in CSF [34].

### 3.5. Genetic Determinants 

Independent replication in CSF and blood compartments have been shown [63,64,65,66], with greater genetic diversity of CSF escape viruses compared to plasma viruses according to phylogenetic analyses of pol/RT [63]. Tong used deep sequencing to explore the range of minority resistance associated mutations in paired CSF and plasma and discovered mutations in CSF that were not identified by Sanger sequencing alone [67]. The evolution of resistance mutations is also seen in CSF viruses [3,30,33,38]. Peluso found CSF viral resistance in patients in whom resistance genotyping was conducted; 6/7 had NRTI mutations, 5/7 patients had PI mutations and 2/7 patients had NNRTI mutations [4]. We hypothesise that CSF escape is also associated with peripheral drug resistance. In Mukherji’s study of the pooled cohort from CHARTER, NNTC and HNRC, the CSF escape cases were combined with all published CSF cases and showed that M184V/I mutations were detected more frequently in the CSF and plasma of patients with escape; 61% (34/56) and 30% (16/55) of samples respectively, compared to participants without escape, where it was only detected in 7% (3/43) of both CSF and plasma samples [34]. APOBEC3F/G mediated hypermutation has been reported to be associated with compartmentalisation in CSF compared with peripheral blood mononuclear cells (PBMCs) and induced drug-resistant mutations in CSF; G73S in protease, M184V and M230I in reverse transcriptase [68].

### 3.6. Neuroimaging

Reported MRI changes in neurosymptomatic CSF escape/discordance are consistent with diffuse white matter signal abnormality (DWMSA) in the dentate nuclei, brainstem, capsular and subcortical white matter [4,31,36,69]. A retrospective analysis of HIV-infected adult patients undergoing diagnostic lumbar puncture at a centre in the United Kingdom found a 10.3 times increased odds of DWMSA in participants with CSF discordance and a 56.9 times increased odds of DWMSA in participants with CSF escape [31]. Perivascular enhancement has also been reported [4,69]. The majority of these demonstrated clinical improvement following ART regimen optimisation. There is little evidence to guide when to re-image. In a case series of 10 patients, MRI improvements tended to lag behind clinical improvement by 2 to 12 months. At 60 days, one patient showed resolution of most focal lesions but the development of a diffuse leukoencephalopathy despite resolution of symptoms. Similarly, another patient had persistent diffuse white matter hyperintensities on MRI at 111 days, with subsequent significant decrease in these abnormalities at 346 and 567 days follow-up [4].

### 3.7. Biomarkers

There are interesting developments in biomarkers of neuropathology and their application to HIV CNS disease [70]. CSF HIV VL correlates well with brain HIV RNA levels but has not been shown to correlate with HAND consistently in some studies [6,71,72]. In primary HIV infection (PHI) when HIV ingresses into the CNS via the blood brain barrier (BBB) or blood CSF barrier (BCSFB), a latent reservoir is established. There is evidence of neuronal injury at this point with elevated CSF neurofilament light chain (NFL) protein and also evidence of macrophage activation with elevated CSF neopterin [73]. In PHI where the BBB is known to be leaky, quotient Albumin (_Q_ALB) (CSF/plasma albumin ratio), a marker of endothelial tight joint permeability is elevated and directly correlates with NFL [73].

Amongst ART treated, neurocognitively unimpaired persons, for most part, CSF neopterin usually decreases to normal levels [73,74,75]. Neopterin remains mostly elevated in those with HAND despite achieving peripheral virological suppression [51,74]. This potentially makes it a useful indicator of HAND, although diagnostic accuracy studies need to be done. NFL may be a useful biomarker for severe NCI. Although it is not disease-specific, it has been found to be mostly normal in those with Alzheimer’s Disease [76] but is largely elevated in those with HAD and untreated neuroasymptomatics with low CD4 counts [77,78]. Although inconclusive due to insufficient power, another study investigating if ANI or MND was associated with neuronal damage found that CSF NFL was higher in individuals with NCI compared with neurocognitively normal individuals but did not predict progressive neurocognitive decline [51]. In patients with HAD, there is also evidence of injury to unmyelinated cortical axons evidenced by elevated total-tau. [79].

The published data on biomarkers in CSF escape/discordance are limited. However, case series have reported evidence of immune activation with elevated neopterin in CSF but not plasma in both symptomatic [4] and asymptomatic cases [26,62] of CSF escape. Markers of BBB dysfunction are evident with elevation of the matrix metalloproteinases MMP-2, 3 and 9 and subsequent elevation of their inhibitors TIMP-1 and 2 [80]. Other inflammatory mediators are elevated such a sCD14—a marker of macrophage activation [80]; chemokines—CCL3, CCL4, CCL5 and CXCL10; cytokines—IL-1α/β, IL-1RA, IL-8, IL-10; TNF related proteins—TNFα, TNFR1, TNFR2 and TRAIL; adhesion molecules; VCAM and ICAM [5,80]. The presence of these inflammatory mediators show that CSF escape/discordance is associated with neuroinflammation, which is occurring in the context of a reconstituted immune system, which may be responsible for both the inflammation and viral replication by attracting HIV infected immune cells or activating latent reservoirs [81]. These inflammatory mediators are significantly higher in the CSF compared to plasma and suggests intrathecal production and that the CNS cellular immune response to viral antigen is compartmentalised [62,80]. NFL has been shown to be elevated in symptomatic CSF escape but slowly normalises as CNS viral suppression is achieved [80]. NFL lends itself to being used as a non-invasive clinical test as plasma NFL, although 50-fold lower, correlates strongly with CSF NFL (r = 0.89, *p* < 0.0001) [77].

### 3.8. Treatment Options

ART as well as being neuroprotective [82], improves neuropsychological functioning and reduces neurological abnormality [83]. There are no evidence-based interventions to guide the best treatment strategies in neurosymptomatic patients with CSF escape/discordance and recommendations are based on expert opinion. In case reports and case series, optimising ART to improve brain penetration has been found to improve symptoms and reduce CSF VL in symptomatic individuals [3,4,35,69,84]. The British HIV Association and European Aids Clinical Society guidelines suggest that in those with a detectable viral load in the CSF, an assessment for CSF HIV resistance should be undertaken and ART optimised to the resistance profile [61]. The MIND exchange program similarly recommends modifying the ART regimen to improve the CPE score and target the CSF viral resistance profile [85].

One published study used 1 g of intravenous methylprednisolone for 5 days followed by a tapering dose of oral prednisolone to treat neurosymptomatic CSF escape/discordance with mixed results [37]. Although neurosymptomatic CSF escape/discordance is accompanied by a degree of compartmentalised neuroinflammation, there are insufficient data to recommend the use of corticosteroids in this context.

## 4. Association between CSF Escape/Discordance and HAND

In the era of ART, the neuropathogenesis of HAND is not simply due to viral replication and neuroinflammation but with multifactorial contribution including the legacy of neuronal injury, HIV-induced metabolic changes, neurotoxicity of ARTs and poor drug penetration [81,86]. Other recognised associations with HAND include comorbid factors such as, drugs and alcohol, liver disease and illiteracy [8]. In ART naïve patients in South Africa, older age and fewer years of education were associated with HAND [21].

A cross-sectional study of the NNTC cohort of post-mortem brain tissue showed that the presence of brain HIV RNA did not correlate well with HAND diagnosed in the absence of features associated with HIV-replication, that is, HIV encephalitis or microglial nodule encephalitis. However, a higher brain HIV RNA level correlated with worse performance on NP tests [6]. This suggests that the mere presence of HIV in the brain is not sufficient to cause neurocognitive dysfunction and that replication with the attendant neuroinflammation may be necessary. It is further support that CSF escape/discordance as a reflection of brain parenchymal HIV replication [6,39], is important in the pathogenesis of HAND.

CSF escape/discordance is reported in patients with HAND. The detection of HIV-1 in CSF even at low-level (<50 copies/mL) using single copy assay was 12.7% in a CHARTER subgroup study [87]. In this cohort, low-level CSF viraemia at any level was correlated with worse neurocognitive performance (r = −0.20, *p* = 0.002) and the presence of CSF escape at first or second visit was associated with a decline in neurocognitive performance compare to those without CSF escape (*p* = 0.02) [87]. In Mukherji’s pooled cohort, symptomatic neurocognitive impairment was more frequent in participants with CSF escape compared to participants without CSF escape (35% vs. 20%) [34].

Although the precise contribution of CSF escape/discordance to HAND is unclear, independent replication of HIV-1 in the brain likely contributes to neurocognitive decline particularly in the context of drug resistance. As CSF escape/discordance is a potentially modifiable risk factor for neurocognitive decline, there does appear to be a role for detecting, quantifying and sequencing CSF HIV RNA in HAND diagnosis and management [61,85,88]. There are no published studies examining the association between CSF escape/discordance and HAND in SSA.

## 5. Research Gaps in Sub-Saharan Africa and Proposed Future Research

### 5.1. Determining the Prevalence of HAND 

Diagnosing HAND has challenges in this setting. It requires a battery of NP tests, that require a specific skill set to deliver and sometimes specialised equipment. These tests were developed in resource-rich settings [13] and until recently normative scores did not exist for resource-limited settings (RLS) [15,18]. Studies compiling normative comparative data in RLS have largely used small convenience samples [14,17,18,22,89]. Two studies have collected large-scale, comprehensive normative neurocognitive data in RLS, using broadly representative participants of the local HIV-negative population and found country specific differences in NP performance [15,90]. This highlights the need for further research to firstly evaluate NP tests in SSA to limit cultural bias and inform on the appropriate NP tests to use for each setting. Secondly, there is a need for comprehensive collection of neurocognitive normative comparison data from HIV-negative controls in SSA.

### 5.2. Pathogenesis of HAND

The biological causes of HAND in ART treated patients in SSA are not well understood however there is evidence to suggest that HIV-1 subtype may play a role [10,11]. Most of the published studies on the pathogenesis of HAND comes from Europe and USA where subtype B HIV-1 predominates. It is therefore important to study the mechanisms driving HAND and CSF escape/discordance in SSA, where the global burden of HIV is greatest and where subtypes C, A, D, G and AG HIV-1 predominate.

### 5.3. CSF Escape/Discordance

There are currently no published data on CSF escape/discordance from SSA. Based on the current knowledge of risk factors for CSF escape/discordance, which include long duration on ART, low nadir CD4 count, LLV, history of plasma viral blips and frequent drug interruptions, together with the rise in the prevalence of pretreatment drug resistance in RLS [91], it is hypothesised that the prevalence of CSF escape/discordance will be higher in SSA compared to data published from resource rich settings. Due to the high prevalence of pretreatment drug resistance in RLS there are imminent plans to change the standard first line ART regimen in SSA from NNRTI-based to dolutegravir-based triple therapy [92], it is important to determine the prevalence of CSF escape/discordance on this new regimen. Going forward, dual therapy may have a role in treatment simplification in virologically suppressed patients in SSA and it would similarly be important to determine the prevalence of CSF escape/discordance on these regimens.

Further research should aim to determine the prevalence, the clinical spectrum and pathophysiology in symptomatic and asymptomatic patients, the genetic variability of compartmentalised viruses, the evolution of drug resistance in this setting, as well as the role of HIV-1 subtypes in the pathogenesis of CSF escape/discordance. Prospective follow up studies are required to determine if asymptomatic persons with CSF escape will progress to symptomatic CSF escape.

### 5.4. Clinical Biomarkers

No studies in SSA have systematically evaluated plasma or CSF biomarkers in ART treated person with CSF escape/discordance. Investigating biomarkers may pave the way to developing useful, non-invasive diagnostic tests for assessing and managing patients with neurological involvement of HIV-1.

## 6. Conclusions

The overwhelming view is that in the era of cART, the neuropathogenesis of HAND is not simply due to viral replication and consequent neuroinflammation but a multifactorial aetiology including the legacy of neuronal injury, HIV-induced metabolic changes, neurotoxicity of ARTs, poor drug penetration, drugs and alcohol and liver disease. CSF escape/discordance does however appear to have an independent role in NCI. Furthermore, the presence of ongoing replication in the CNS may be a barrier to HIV-1 eradication and the goal of achieving a cure.

## Figures and Tables

**Table 1 brainsci-08-00190-t001:** Definition of CSF escape and CSF discordance. * Depending on the limit of quantification of the assay used. Usually 50 copies/mL. VL; Viral Load.

	Plasma VL	CSF VL
**CSF escape**	Undetectable *	Detectable *
**CSF discordance**	Detectable *	Greater than 0.5 or 1log_10_ of the plasma VL

**Table 2 brainsci-08-00190-t002:** Frascati Criteria for diagnosing HIV Associated neurocognitive disorder. ADLs; activities of daily living. SD; standard deviation.

Asymptomatic Neurocognitive Impairment (ANI)	Mild Neurocognitive Disorder (MND)	HIV-Associated Dementia (HAD)
No interference with ADLs	At least mild interference with ADLs	Marked interference with ADLs
At least 1.0 SD below mean of normative population in at least two cognitive domains	At least 1.0 SD below mean of normative population in at least two cognitive domains	At least 2.0 SD below mean of normative population in at least two cognitive domains

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
