# Peer review of "HIV Cerebrospinal Fluid Escape and Neurocognitive Pathology in the Era of Combined Antiretroviral Therapy: What Lies Beneath the Tip of the Iceberg in Sub-Saharan Africa?"

_brainsci, 2018, doi:10.3390/brainsci8100190_

Round 1

Reviewer 1 Report

This is a well written review of CSF escape. Collier and colleagues should be commended for thinking about the the problem and research needs of Sub-Saharan African where it is likely there is a large burden of CSF escape and cognitive impairment.

Some comments/suggestions:

The prevalence of CSF escape in well-treated patients is uncertain but may be lower than the ~10% quoted. However, it should be noted that the prevalence is likely to be influenced by the lower limit of detection of the CSF assay and sample volume. Two recent reports in virologically suppressed patients from the COBRA and CHARTER neuroimaging studies reported the prevalence of CSF escape to be ~1.5%. Some reports quoted are from patients having LPs for clinical reasons or because they had symptoms, which is likely to bias the prevalence. 

Van Zoest et al. J Infect Dis. 2017 Dec 27;217(1):69-81. doi: 10.1093/infdis/jix553.

Underwood, J et al (2018). Journal of Acquired Immune Deficiency Syndromes78(4), 429–436. http://doi.org/10.1097/QAI.0000000000001687

It’s probably worth emphasising that CSF escape is a potentially modifiable risk factor for cognitive decline, unlike nadir CD4, age etc. It’s also important to mention that its important to determine the prevalence of CSF escape with 2nd generation integrate inhibitors like dolutegravir and bictegravir given they are likely to be important drugs in naive and treatment experienced patients going forward. Similarly, quantifying the prevalence of CSF escape in new treatment strategies like dual therapy (e.g. 3TC/PI, 3TC/DTG, DTG/RPV) is important going forward. This is also relevant to SSA as dual therapy may be a viable way of reducing cost in an antiretroviral programme provided it is efficacious.

Perhaps consider revising pathogenesis/risk factors into viral, host and treatment sections. Additionally, the section on the association of CSF escape and HAND should probably be earlier in the manuscript as it overlaps with the clinical presentation part.

Regarding normative data, having study-specific HIV-uninfected controls is the best way to assess the true burden of HIV-associated cognitive impairment in SSA.

Some minor points:

Line 53 - define SSA here not in 55

Line 67 - the prevalence of actual pathology is debatable due to methodological flaws of the HAND criteria - see Underwood et al 2018 and Gisslen et al for modelling showing the expected rate of cognitive impairment in a normally distributed population (i.e. the false positive rate)

Line 76 - I’m not sure this is the ‘gold standard’ or if there even is one. This is a research definition and was not intended for use in clinical practice. Consider revising.

Line 148 - These

Line 150 - Is it not more likely to be more of a continuum and we don’t really understand the pathophysiology particularly well? Symptoms can be vague and the correlation between them an objective markers of cognitive impairment are not consistent.

Line 181 - or often to prevent toxicity from TDF or ABC

Line 202 - where

Line 248 (and another instance) - consider revising ‘respectably’. Is it really disrespectful not to have a certain degree of immune reconstitution? Escape occurring with a more a active immune system may be associated with a higher concentrations of biomarkers or a different phenotype of immune activation.

Line 256 - elevated phosphorylated tau is a biomarker of AD. It’s not a biomarker for ageing. Using machine learning and neuroimaging data it has been shown that even those with well treated HIV have a degree, albeit modest, of premature ageing. Anyhow, this paragraph doesn’t seem related to CSF escape. Consider removing.

Cole et al. Neurology. 2017 Apr 4;88(14):1349-1357.

Underwood, J et al. Journal of Acquired Immune Deficiency Syndromes, 2018; 78(4), 429–436. http://doi.org/10.1097/QAI.0000000000001687

Line 307 - define RLS. 

Line 339 - drugs

Line 341 - may be. 

Reviewer 2 Report

Collier et al. discuss in this review six important points regarding the role of HIC CSF escape/discordance and the development of HIV-associated neurocognitive disorders (HAND) in the era of cART: 1) the prevalence of HAND globally, 2) the clinical spectrum of CSF escape/discordance 3) the contribution CSF escape/discordance to HAND, 4) critically review the role of CSF HIV RNA and the evolution of HIV drug resistance in the pathogenesis of CSF escape/discordance, 5) appraise the current state of clinical biomarkers in the diagnosis of HAND and CSF escape/discordance and 6) review progress with regard to disease management. A point of major importance in this review is that data speaking to many aspects discussed here are extremely scarce for sub-Saharan Africa (SSA) where the HIV epidemic has hit the worst. The need for scientific work and clinical translation in SSA can hardly be overemphasized, and thus a review like this one is not only very timely but in a way long overdue.

As of the manuscript the following points need to be addressed for clarification: Introduction, the last paragraph contains partially redundant sentences.

Page 4, lines 136-146, pathogenesis – HIV infection of macrophages has been well documented in the literature and it is notable that even small numbers of infected monocyte-macrophages found in the periphery have significant effects on the development of HAND (e.g. see Valcour et al.). The difficulty of pathologists to find in the era of cART in a strongly reduced number of total autopsies HIV-infected multinucleated giant cells cannot be taken as evidence that macrophages/microglia are not infected. Neither does the ability of macrophages to phagocytose HIV-infected CD4+ lymphocytes provide evidence against direct infection of macrophages. The phagocytosis phenomenon may just be easier to detect. Besides, it has been well documented that macrophages/microglia do not need to be infected but just be exposed to HIV components in order to produce neuronal toxins, injury and eventually neuronal cell death.

Page 6, line 256 - the first sentence is confusing with regard to its statement on aging. There is accumulating evidence in the literature that HIV-infected individuals show signs of deteriorating health and frailty associated with aging about 10 to 15 years earlier compared to un-infected persons. Whether or not CSF escape/discordance affects aging–related neurocognitive deterioration remains to be elucidated.

Reviewer 3 Report

The authors have written a timely, useful and relatively concise review on the topic of HIV CSF escape, discussing what is known from studies conducted in Western settings and highlighting some of the barriers to making progress in this area in sub-Saharan Africa. 

There is a typo in line 238. 
